# The Role of GPR15 Function in Blood and Vasculature

**DOI:** 10.3390/ijms221910824

**Published:** 2021-10-06

**Authors:** Mario Bauer

**Affiliations:** Department of Environmental Immunology, Helmholtz Centre for Environmental Research—UFZ, 04318 Leipzig, Germany; mario.bauer@ufz.de

**Keywords:** GPR15, C10orf99, thrombomodulin, lymphocyte, blood, endothelial cell, vasculature

## Abstract

Since the first prominent description of the orphan G protein-coupled receptor 15 (GPR15) on lymphocytes as a co-receptor for the human immunodeficiency virus (HIV) type 1 and 2 and the first report about the GPR15-triggered cytoprotective effect on vascular endothelial cells by recombinant human thrombomodulin, several decades passed before the GPR15 has been recently deorphanized. Because of new findings on GPR15, this review will summarize the consequences of GPR15 signaling considering the variety of GPR15-expressing cell types and of GPR15 ligands, with a focus on blood and vasculature.

## 1. GPR15

GPR15 is a member of the Class A (rhodopsin) orphan G protein-coupled receptor (GPCR) family. It is expressed in high abundance in colonic mucosa, small bowel mucosa, and the liver, and is expressed in low abundance in lymph nodes, testes, and prostate [1]. In blood vessels, it is normally expressed on endothelial cells (ECs). In contrast to the vessel wall, its expression on blood cells is dependent on both cell type and cell subtype. It is mainly expressed on T and B lymphocytes but not on monocytes and neutrophils [2]. The majority of GPR15+ T cells are central memory (>80%) T cells. GPR15+ T cells are found in different subtypes of mainly T-helper cells (Th, >50%). The surface expression of GPR15 on T-helper cells (Th) appears to be unique, not concomitantly accompanied by the expression of another protein [3]. Interestingly, the frequency of GPR15+ T cells in blood serves as a biomarker for chronic cigarette smoking independently of cell subtype, as to date, no other stressor affecting GPR15 has been found. It has been assumed that smoking instead influences the imprinting of T cells into GPR15+ T cells in lymphatic organs, as cigarette smoke extract cannot affect either the cellular GPR15 expression level or the frequency of GPR15+ T cells in vitro [2]. In contrast to lymphocytes, no reports of affected GPR15 expression in EC have been identified to date. More details about the biological effects of smoking and GPR15 activation are described in a former review [4].

Under pathophysiological conditions, GPR15 appears to constitute a counter-regulator of inflammation. Knocking out GPR15 in mice led to an aggravation of inflammation in skin for autoantibody-mediated disease [5] or in the colon for an anti-CD40 antibody model of innate immune cell-mediated colitis [6]. However, there are additional reports ruling out an opposing disease-promoting effect and species-dependent differences indicating a more complex role of GPR15 [6].

Since the first prominent description of GPR15 on lymphocytes as a co-receptor for the human immunodeficiency virus type 1 and 2 (HIV) in 1997 [7] and the first report about the successful antithrombotic effect of recombinant human thrombomodulin in 1990 [8], several decades passed before a ligand for GPR15 was found. In 2017, two ligands were discovered, which are clearly different from each other [9,10].

## 2. Endogenous GPR15 Ligands and Binding Protein 

The first activating binding partner for GPR15 was found to be exogenous particles in the form of viruses (Figure 1 and Figure 2). In particular, many simian immunodeficiency virus (SIV) and HIV-2 strains and HIV-1 strains to a lesser extent are able to use GPR15 as an alternative co-receptor to enter cells [11]. The interaction between GPR15 and virus is performed by the envelope protein gp120, particularly with its variable loop 3 (V3) region [12]. Several single amino acid substitutions in V3 completely abrogated the infectivity of mutant virus in GPR15+ cells, indicating the specificity of V3 for GPR15 binding.

With respect to endogenous binding partners, to date, there have been reports describing two different GPR15-activating ligands: one ligand of physiological origin, named C10orf99, most highly expressed in the colon [13], and the other ligand of synthesized origin (recombinant human soluble thrombomodulin, ART-123) applied as an anticoagulant [8] (Figure 1 and Figure 2). In addition to these ligands, a cystatin C fragment in blood was found to inhibit the entry of GPR15-tropic viruses by binding but not activating GPR15 [14] (Figure 1). 

Surprisingly, there are no homologies in protein sequences between the four reported GPR15 binding proteins (Table 1).

**Table 1 ijms-22-10824-t001:** Sequences of GPR15-interacting proteins/peptides. Color code for amino acid residues: green, hydrophobic uncharged; red, acidic; blue, basic; black, other; underline, signal peptide; black boxes, residues essential for binding to GPR15; bluish colored background, cysteine residue.

	Amino Acid Residues	Reference
**GPR15 activating protein (ligand)**		
*C10orf99 (GPR15L)*	MRLLVLSSLLCILLLCFSIFSTEGKRRPAKAWSGRRTRLCCHRVPSPNSTNLKGHHVRLCKPCKLEPEPRLWVVPGALPQV	[15]
*TME5*	QMFCNQTACPADCDPNTQASCECPEGYILDDGFIC	[16]
*TME5C*	ECPEGYILDDGFICTDIDE	[16]
*gp120 V3(HIV)*	CTRPNNNTRKGVHIGPEKVYF TTSIIGDIRQAHC	[12]
**GPR15 binding protein**		
*CysC95-146*	GRTTCTKTQPNLDNCPFHDQPHLKRKAFCSFQIYAVPWQGTMTLSKSTCQDA	[14]

**Figure 1 ijms-22-10824-f001:**
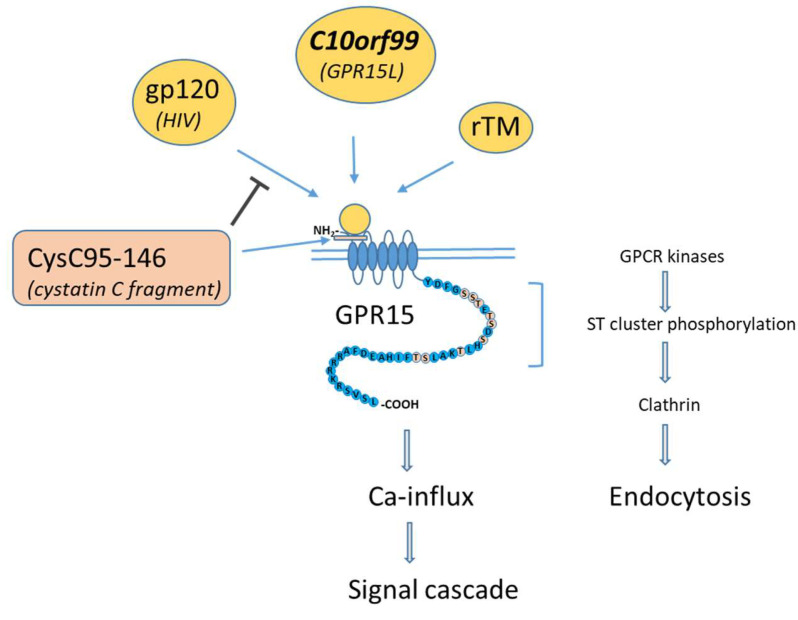
Protein interaction with GPR15, adapted to Okamoto et al., 2017 [17]. GPR15-activating ligands (yellow circles) bind mainly at the extracellular N terminus and to the first extracellular loop (ECL1) of GPR15. The binding protein CysC95-146 is unable to activate GPR15 but can inhibit binding to protein (gp120) of the GPR15-tropic human immunodeficiency virus (HIV). Conformational changes of ligand-bound GPR15 leads to Ca-influx and ligand-dependent endocytosis. Endocytosis is evoked by clathrin-activating phosphorylation of the ST cluster of the C-terminus of GPR15 by GPCR kinases.

**Figure 2 ijms-22-10824-f002:**
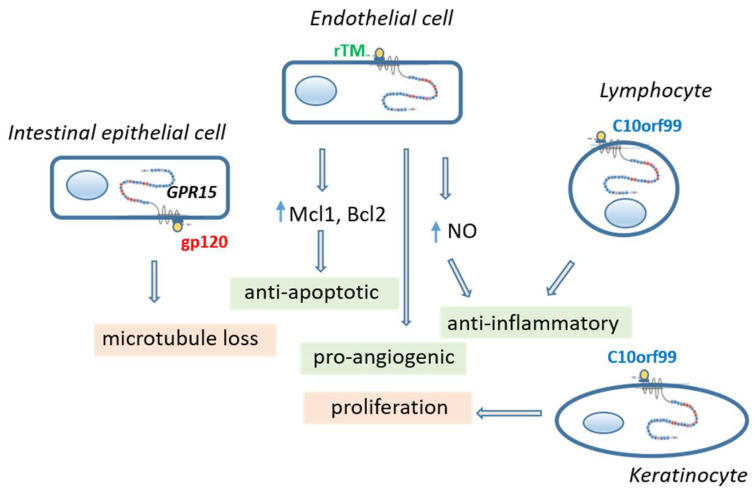
Consequences of GPR15 activation in different cell types by different agonists. Activation of GPR15 in vascular endothelial cell and lymphocytes leads to a greater cell-protecting effect. In contrast, activation of the intestinal epithelial cells by gp120 of HIV causes leakage in the epithelial layer and activation of keratinocytes by C10orf99 causes adverse proliferation. *C10orf99* chromosome 10 open reading frame 99, *gp120* glycoprotein 120 of human immunodeficiency virus, *GPR15* orphan G protein-coupled receptor 15, *Mcl1/Bcl2* anti-apoptotic proteins, *NO* nitric oxide, r*TM* recombinant thrombomodulin.

### 2.1. Physiological Ligand Found in Colon—C10orf99 (GPR15L)

On the basis on the specific homing of GPR15+ lymphocytes into the colon, it was proposed that ligands could be found in this specific tissue. One natural ligand was identified in 2017 from porcine colon extracts in an in vitro model using human GPR15-expressing Chinese hamster ovary (CHO) cells engineered to express the promiscuous G protein Gα_16_ to follow ligand binding by means of calcium release [10]. The human transcript orthologue of this identified pig protein was annotated to the chromosome 10 open reading frame 99 gene, *C10orf99*. 

C10orf99 encodes for a short basic amphiphilic secreted peptide (GPR15L) of 81 amino acid residues initially called “9 kilodalton CC-motif containing cationic polypeptide AP57/colon-derived sushi containing domain-2 binding factor (CSBF)”. Because of its protein characteristics and features, it is proposed that this is a new type of multifunctional antimicrobial peptide. It is found mainly on mucosal and skin epithelium as well as in some tumor and/or their adjacent tissues, such as esophageal cancer, hepatocellular carcinoma, squamous cell carcinoma, and invasive ductal carcinoma [15]. Additionally, it is also present in low abundance in blood or cerebrospinal fluid [18]. The general physiological role of C10orf99 is not well understood but seems to be tissue dependent. In mucosal epithelium, C10orf99 is more constitutively expressed and less affected by inflammation or the presence of microbiota [19], although there is a strong specific homing of GPR15-expressing T cells mainly into the colon.

In contrast, C10orf99 (GPR15L) was upregulated in skin in all examined inflammatory diseases, including psoriasis, atopic dermatitis, contact eczema, and *lichen planus* [9,20]. It must be checked on a case-by-case basis whether C10orf99 primarily acts via GPR15. Despite the induction of C10orf99 in psoriasiform dermatitis in mice, GPR15+ cells did not accumulate in the skin, which led to the proposal of a more GPR15-independent pathway by the author [21]. In contrast, C10orf99 was considered to act in a GPR15-dependent pathway on keratinocytes in psoriasis [22].

Recently, it has been shown that post-translational modifications (PTMs) of the GPR15 receptor, such as sulfated N-terminal tyrosine residue(s) or the disruption of O-glycosylation on the N-terminal threonine or serine residues, or the removal of α2,3-linked sialic acids from O-glycans, enhances binding to C10orf99 [23]. For C10orf99 activity, its extreme C-terminal residue and its hydrophobicity were considered to be necessary for optimal receptor–ligand interaction.

### 2.2. The Synthesized Ligand Applied as Anticoagulant—Thrombomodulin

Analyzing the cause of the cytoprotective effect of the recombinant human soluble thrombomodulin (rTM, ART-123) on endothelial cells, in the same year, in 2017, an additional ligand for GPR15 was found [9]. The rTM has been used as an anticoagulant to treat patients undergoing disseminated intravascular coagulation (DIC) [24]. Specifically, immunoprecipitation of membrane proteins of human umbilical vein endothelial cells (HUVECs) with the fifth epidermal growth factor-like region (TME5) of TM revealed GPR15 as a binding partner. Similar binding effects of this short TME5 (35 amino acids residues) have also been found for the greater rTM, suggesting E5 of TM as the main binding region for GPR15. More specifically, the third loop of TME5, called TME5C, composed of 19 amino acids activates GPR15 signaling [16]. Comparing the protein sequence of the 19 amino acids of TME5C or 35 amino acids of TME5 with that of the 81 amino acids of C10orf99, homologies between both ligands are surprisingly absent (Table 1). Homologies with the most preserved amino acids of C10orf99 from different species have also not been found for the TME5C sequence.

Apart from the rTM, heterogeneous soluble TM fragments from vascular endothelial cells circulate in the plasma and are found at increased levels in various diseases, such as cardiovascular disease and diabetes, and in ischemic and/or inflammatory endothelial injuries [25]. It has been shown that these soluble TM fragments retain significant anticoagulant activities. Since the fourth and fifth regions of an EGF-like domain (TME45) act as an anticoagulant by binding thrombin [26], it can be assumed that these TM fragments may have effects on endothelial cells via the GPR15 receptor.

### 2.3. The GPR15 Binding Protein—Cystatin C Fragment

In 2021, a C-terminal fragment of cystatin C (named CysC95-146) but not the full-length cystatin C was identified to bind to GPR15 [14] by screening a hemofiltrate (HF)-derived peptide library containing peptides and small proteins circulating in human blood in their final processed and physiologically relevant forms. It has been shown that this fragment inhibits the entry of GPR15-tropic derivatives of HIV and SIV in human osteosarcoma (GHOST) cells that stably express CD4 and GPR15. Virus entry is indicated in ghost cells by a green fluorescent protein (GFP) gene that is under the control of the virus. Similarly, in peripheral blood mononuclear cells (PBMCs) stimulated with PHA and IL-2 to enhance the frequency of GPR15+ cells, this inhibition of virus entry was replicated but is more pronounced for SIV compared to GPR15-tropic derivatives of HIV-1 or HIV-2. Interestingly, the GPR15 ligand C10orf99 did not affect the inhibitory effect of CysC95-146 in PBMCs, which indicates different binding sites. Competing experiments with anti-GPR15 antibodies revealed that CysC95-146 binds to the extracellular N terminus and to the first extracellular loop (ECL1) of GPR15. Binding to these GPR15 regions prevents viral entry but does not activate the GPR15 receptor and does not inhibit binding of C10orf99. Essential binding sites were found by amino acid exchanges at positions G69A, K94A, and Q100A, which completely abrogated inhibition of viral entry by CysC95-146.

Cystatin C is an inhibitor of cysteine proteinases consisting of 146 amino acids. It is found in nearly all cells with a nucleus [27] and appears to be one of the most important extracellular inhibitors to prevent the breakdown of proteins [28]. In blood, it serves as a marker of glomerular filtration. The reference interval for plasma ranges from about 0.58–1.00 mg/L in women and 0.62–1.04 mg/L in men [29]. Reference values for the cystatin C fragments do not exist, and as a result, their physiological role has not yet been determined. Compared to blood, CysC95-146 was found at a much lower concentration of about 0.01 mg/L in the diluted hemofiltrate. An extrapolation to the plasma level cannot be carried out due to severe differences between blood and hemofiltrate. However, a half-maximal inhibitory concentration (IC_50_) for viral entry of about 25 mg/L (0.5 µM) has been described for CysC95-146. By greatly exceeding the normal range of the parental cystatin C in blood, GPR15-binding activities for cystatin C fragments in blood appear to be less likely. As a consequence, evidence for the active role in binding GPR15 of these fragments still needs to be provided.

## 3. Physiological Role of GPR15 Ligands in Vascular Tissue 

The different physiological role of the two GPR15 ligands, C10orf99 and thrombomodulin species, for lymphocytes and endothelial cells is illustrated and described in more detail in Figure 3.

### 3.1. C10orf99

Although C10orf99 can normally be found in plasma in low concentrations (1-6 ng/mL, manufacture notes), its major role appears to be to attract GPR15-bearing blood cells from the outside of vessels, such as the colon and skin, as mentioned above. Apart from GPR15, however, C10orf99 may act as a tumor suppressor by suppressing proliferation of several tumor cell lines via G1 arrest by interacting with another binding receptor, called sushi domain containing 2 (SUSD2) [13]. Nevertheless, its role for GPR15 signaling on vascular ECs or on attracted lymphocytes has not yet been described in detail.

### 3.2. Thrombomodulin Peptides

The cellular consequences of GPR15 activation on ECs were exclusively described for the rTM, in particular for TME5 and its C-loop TME5C, but not for the natural ligand C10orf99. By supposing soluble TM fragments as a putative natural ligand of GPR15 on EC, it has been shown that this receptor–ligand linkage mediates both cytoprotection and pro-angiogenic activity on ECs [3,16,31]. Cytoprotection was defined as the attenuation of growth inhibition and apoptosis caused by the calcineurin inhibitor FK506 or cyclosporinA (CsA). It has been shown that TME5 and TME5C induce activation of extracellular signal-regulated kinase (ERK) (p-ERK) and AKT serine/threonine kinase 1 (p-Akt) in human umbilical vein ECs (HUVECs), leading to upregulation of the anti-apoptotic myeloid cell leukemia sequence 1 (Mcl-1) protein. As a consequence, TME5 and TME5C could block calcineurin inhibitor-induced capillary leakage. The effect of TME5 or TME5C was exclusively mediated by GPR15, as cytoprotective and pro-angiogenic effects of these TM species were absent in ECs from *Gpr15* knock-out mice.

The cellular consequences of GPR15 activation on circulating lymphocytes in human blood has not been studied to a sufficient extent. However, the first indications of a greater anti-inflammatory effect of TME5 have been described in mice [32,33]. It has been shown that TME5 alleviated murine graft-versus-host disease [32] or LPS-induced sepsis [33] in a GPR15-dependent manner. Anti-inflammatory signs were given by the TME5-induced increase in the number of induced regulatory T cells (iTreg) in a mixed lymphocyte reaction, by suppressed upregulation of pro-inflammatory IL-6 in association with an inhibited NF-kB pathway in activated T cells or a reduced activation of dendritic cells. Thus, in addition to the anti-inflammatory effect of thrombomodulin through the lectin-like domain [34], the C-loop of the fifth region of the EGF-like domain of TM (TME5C) preserves anti-inflammatory activity through the GPR15 receptor.

Interestingly, besides constitutive expression of TM in ECs, TM is also found at low expression levels in monocytes. Under certain pathological conditions, such as the inflammatory environment in the bone marrow of patients with low-risk myelodysplastic syndromes (MDS), TM is overexpressed, especially in classical monocytes in the bone marrow but also in peripheral blood [35]. The anti-inflammatory effect of TM was confirmed by induction of a more anti-inflammatory profile of CD4+ T cells in the presence of TM+ monocytes compared to TM- monocytes.

## 4. Influences on GPR15 Expression

In considering GPR15 as a novel therapeutic target to cure immune disorders or vascular diseases it is important to elucidate mechanisms that control surface expression. In the following discussion, data on receptor endocytosis, proposed enhancers or inhibitors, the influences of DNA methylation, and impaired expression under cellular transformation will be reviewed.

### 4.1. Endocytosis

Because the cell surface density of GPCRs can be modulated by endocytosis [36], it has been shown specifically for GPR15 that it can undergo both ligand-dependent endocytosis and constitutive endocytosis in the absence of ligand [17]. Constitutive endocytosis in the absence of ligand was independent of cell type. GPR15 internalization was demonstrated for both adherent cells (HEK293) and non-adherent lymphoblast cell lines. The internalization rate ranged from about 10 to 30%. It has been stated that endocytosis requires a constitutive phosphorylation of Ser-357 in the distal C-terminus, which could be induced by members of the kinase family, such as PKA, PKC, and AKT. The PKC activator phorbol 12-myristate 13-acetate (PMA) additionally promoted endocytosis up to 55%, which suggests an additional impact of PKC on the endocytotic machinery. In contrast with ligand-induced endocytosis of GPCRs, constitutive endocytosis is only partially dependent on the β-arrestin-activated endocytic coating protein clathrin. The endocytosis of GPR15 seems to be accompanied by receptor degradation, since only about half of the endocytosed GPR15 recycled to the plasma membrane. Using C10orf99 as a GPR15 ligand, a convincing ligand-specific endocytosis has been documented by achieving almost complete endocytosis of ligand-activated receptors in the GHOST-GPR15 cell line [14]. In contrast to constitutive endocytosis, this endocytosis mechanism is mainly activated by conformational changes of the receptor leading to phosphorylation of the ST cluster at the distal C-terminus by GPCR kinases, but it does not need the phosphorylation of Ser-357 [17].

In conclusion, one mechanism affecting the density of GPR15 at the surface of GPR15-expressing cells is phosphorylation-regulated and ligand-independent constitutive endocytosis. This means that all endogenous or exogenous factors influencing the constitutive phosphorylation of Ser-357 would affect GPR15 expression. Finally, the evidence of ligand-independent endocytosis of GPR15 could serve as the basis for a strategy to develop therapeutics in the form of GPR15 antibodies coupled with therapeutic agent, as has been shown for the transferrin receptor [37].

### 4.2. Enhancers

Enhancers of GPR15 expression have mostly been described for lymphocytes. However, reported effects should be treated with caution, as a distinction must be made between upregulation of GPR15 at the cellular level and upregulation of the frequency of GPR15+ cells within a cell population.

#### 4.2.1. Enhancers—Upregulation of the Frequency of GPR15+ Cells

Upregulation of GPR15+ cells has been evoked in human PBMCs stimulated with phytohaemagglutinin (PHA) and IL-2 for three days [14]. Because of the mitotic activity of PHA on lymphocytes and monocytes and the proliferative activity of IL-2 on lymphocytes, it cannot be excluded that this upregulation was an upregulation of the frequency of GPR15+ lymphocytes rather than a de novo upregulation of GPR15 in single lymphocytes. A surprisingly almost identical finding was found when mice T cells were stimulated with anti-CD28, anti-CD3, and IL-2 for 3 days [33]. In both studies, the frequency of GPR15+ lymphocytes increased about threefold (~4 to ~14%). The presumption that GPR15-expressing cells have an advantage over other lymphocytes in the case of mitotic stimulation has not been proven but could be determined by a more anti-apoptotic state of GPR15+ cells in the case of ligand activation. Interestingly, an activation of GPR15 receptors by the ligand TME5 attenuated proliferation of activated spleno-T cells by irradiated allo-geneic splenocytes in a mixed lymphocyte reaction (MLR) [32]. Inhibition in proliferation was apparent for the T cell subtypes Th1 and Th17 cells but not for induced T-regulatory cells (iTreg), where an increase in cell count was found. The reason for this strong effect of TME5 on the proliferation of activated T cells is not yet clear, considering that the frequency of GPR15+ T cells in the in vitro culture systems does not exceed 15%. Supposing an exclusive binding of TME5 to GPR15, an anti-proliferative effect on GPR15-activated T cells could be proposed. However, this assumption opposes to some extent the angiogenic effect of GPR15 activation on ECs.

In blood, the most prominent upregulation of *GPR15* in PBMCs was found to be associated with chronic cigarette smoking [3,18,38,39,40]. This upregulation slowly reverses over many years after cessation to the range of never smokers. The reason for the increased *GPR15* gene expression was based on an increased frequency of GPR15+ T cells, as the main fraction of GPR15+ cells in blood, and not by an upregulation of *GPR15* in given T cells [4].

#### 4.2.2. Enhancers—Upregulation of GPR15 Expression

Upregulation of GPR15 expression could be caused by exogenous or by endogenous factors. With respect to exogenous factors, the comparative toxicogenomics database (www.ctdbase.org, accessed on 13 August 2021) lists nine interacting chemicals resulting in increased *GPR15* mRNA expression and three chemicals resulting in decreased mRNA expression. With respect to human blood and vasculature, there was only one report showing only a marginal increase of *GPR15* (<1.5-fold) in T cells exposed to benz(a)pyrene in vitro [41]. Another much stronger inducer of GPR15 expression, especially in lymphocytes, is the Epstein–Barr virus (EBV). The mechanism behind this remains elusive, but according to the GTEx portal (www.gtexportal.org, accessed on 13 August 2021), EBV-transformed lymphocytes show by far the strongest expression compared to other diverse human tissues.

With respect to endogenous conditions, an interesting finding was the strong upregulation of GPR15 expression at the protein level in synovial macrophages of chronically inflamed joints in the case of rheumatoid arthritis (RA) [42]. This suggests a more chronically disturbed homeostasis affecting the GPR15 expression level. Remarkably, there was a strong interindividual variance in *GPR15* expression in rheumatoid synovia. An upregulation was present only in 25% of patients with RA, which makes it difficult to draw a more generalized conclusion.

The findings of an increased frequency of GPR15+ lymphocytes in the blood of smokers but the absence of induction of GPR15 by smoking compounds led to the assumption that at least two factors could be responsible for GPR15 expression: one factor that triggers proliferation, and a second factor that imprints lymphocytes to express GPR15 in lymph nodes where the GPR15+ lymphocytes come from. Particularly interesting in this regard is the observation of a strong increase in GPR15+ cells among proliferating CD4+ T cells stimulated in vitro for several days with heat-killed candida albicans [18]. This effect was not found in non-proliferating cells or in other tested stimulants and was independent of smoking.

Because of the limited investigations on the interactions of transcription factors with GPR15 promoters and enhancers so far, greater efforts are needed to clarify the mechanisms of GPR15 expression. Although initial evidence shows the role of GATA3 or FOXP3 binding to GPR15 enhancers upstream of *GPR15* leading to propagation of GPR15 in Th2 or repression of GPR15 in Treg, respectively [43], it does not explain the presence and, in particular, the smoking-induced abundance of GPR15+ cells among different T cell subtypes.

### 4.3. Inhibitors

In colorectal cancer (CRC), *GPR15* expression correlated negatively with the expression of microRNA (miRNA) miR-182-3-p, indicating potential inhibition of GPR15 expression by epigenetic mechanisms [44].

Additional potential antagonists for GPR15 were identified using the in silico utilizing virtual screening technique [45]. At least eight compounds from a chemical database of 62,500 small molecules were identified that putatively antagonize GPR15. Their functional influence has not yet been proven.

### 4.4. DNA Methylation

GPR15 expression appears to be strongly influenced by DNA methylation within the GPR15 gene. It has been shown that GPR15 expression in blood was apparent exclusively in lymphocytes (T and B cells) with approximately 50% methylation at CpG cg19859270 located within the single exon of GPR15 [3]. Non-GPR15-expressing cells are nearly 100% methylated at this site. In addition, excluding imprinting mechanisms, a general random monoallelic expression of GPR15 could be suggested based on complete demethylation of cg19859270 in at least one allele. Such hypomethylation was not observed in myeloid cells, such as granulocytes or monocytes [46,47]. Notably, the differentiation of monocytes into tissue macrophages can evoke GPR15 expression as has been shown for synovial macrophages [42]. It has yet to be shown whether the differentiation-induced GPR15 expression correlates with DNA methylation. Analyzing all nine CpGs of the GPR15 gene body, the hypomethylation of three additional CpGs was found to be associated with *GPR15* expression [48]. Remarkably, this study was based on observations of monocytes, for which hypomethylation in blood would not have been expected. Thus, it appears more likely that these data arose from the non-monocytic cells still present after applied enrichment of monocytes to a purity of about 72% to 85%.

Over the last decade, many population-based studies have been conducted on blood to uncover epigenetic, particularly DNA methylation, signs for diverse pathophysiological conditions. Surprisingly, the strongest association to GPR15 methylation in blood cells was related to smoking, particularly to chronic cigarette tobacco smoking as reviewed in [49]. In addition to cigarette smoking, cannabis smoking similarly mediates hypomethylation at cg19859270 in cells of the adaptive immune system accompanied by an excess in GPR15+ Th cells [39]. Whether combustion products from electronic cigarette smoking would exert a similar effect has not been studied to date. The consideration of methylation at cg19859270 in whole blood as a biomarker for chronic smoking was less successful at the subject level because of small methylation differences (<3%) between smokers and never smokers. In contrast, analysis of the frequency of GPR15+ T cells serves as a much better biomarker in blood for chronic smoking at the subject level [50].

### 4.5. Degeneration of Cells

The finding that C10orf99 suppresses proliferation in tumor cells via SUSD2 [13] led to the hypothesis that similar effects could be evoked via GPR15 in cancer. To prevent such anti-proliferative effects via GPR15, tumor cells would downregulate this receptor. Indeed, analyzing publicly available data from Cancer Genome Atlas (TCGA) and the Genotype-Tissue Expression (GTEx) databases, among 33 cancer types, *GPR15* expression was significantly lower in only two cancer types, the colon and rectal adenocarcinoma (COAD and READ respectively), compared to healthy tissue [44,45]. Although the cellular distribution of GPR15 in cancer or healthy tissues could not be specified, the downregulation of *GPR15* expression, especially in the colon, could also be the result of a physiological excess of GPR15+ lymphocytes in healthy colon tissue into which these cell types specifically home. However, colorectal cancer in progressed stages could be additionally infiltrated by tumor-associated GPR15+ Tregs of the Th17-like phenotype. In this microenvironment, these cells prevent CD8+ T cell-mediated antitumoral immunity [51].

Looking for interaction partner or co-expressed genes, integrated network analysis confirmed the physical interaction to *YWHAB* encoding of the protein 14-3-3 [45] that binds to the C terminal of GPR15 to promote its cell surface expression and to increase its stability [52]. Putative *GPR15* co-expressed genes, such as *TACR1*, *TAS2R9*, *SPDYE4*, or *GPR182*, could be a more specific sign for tumor cells of adenocarcinoma or for healthy epithelial cells from which the tumor originates since all four genes were not expressed in GPR15+CD3+ lymphocytes [4].

In contrast, another recent report showed an upregulation of GPR15 in colorectal cancer in comparison to normal adjacent tissue [53]. Repression of GPR15 strongly induced apoptosis and inhibited colorectal cancer cell growth, migration, and invasion.

## 5. Open Questions Relating to GPR15 Receptor Binding in Blood and Vasculature

(i).In contrast to large rTM, such as ART-123, TME5 has no anticoagulant activity by failing to bind thrombin. Nevertheless, it might be of interest how the affinity of rTM to thrombin or GPR15 influences rTM binding patterns to these two binding proteins in blood vessels.(ii).It also remains to be determined whether natural soluble TM fragments are capable of affecting endothelial cells via their GPR15 receptor and whether soluble TM fragments are the most prominent natural ligands of GPR15 for ECs.(iii).Still unresolved is the effect of rTM when used as an antithrombotic drug for the physiological function of GPR15+ lymphocytes. It should be shown that rTM would not impede the homing of these lymphocytes into other tissues in particular into the colon and, as a consequence, would not affect tissue homeostasis in the colon.(iv).The role and effect of C10orf99 on endothelial cells has not yet been described. A putative physiological effect on ECs, however, could be expected at sites with microbial contact, such as injured epidermis, colon, oral cavity, and esophagus, where C10orf99 is secreted to a greater extent by epithelial cells.(v).Additionally open is whether GPR15+ lymphocytes bind to the TM at the lumen surface of endothelial cells or to thrombomodulin-expressing monocytes in blood.(vi).Especially for subjects with a higher risk of acquiring vascular diseases, such as atherosclerosis, it is certainly of interest whether the cytoprotective effect of GPR15 activation in ECs can be used to prevent disease-promoting endothelial dysfunction.

## 6. Conclusions

Over the past few years, there has been a renaissance of interest in GPR15. This was presumably triggered by the (i) detection of a causal relationship between antithrombotic thrombomodulin fragments and the cytoprotective effect on vascular endothelial cells, (ii) by the inhibition of tumor growth by silencing of GPR15 [53], (iii) by its role in skin diseases, and (iv) by its role for homeostasis in the colon, the organ interacting with the majority of intestinal microbiota.

The recent evidence of two endogenous but distinctive GPR15 receptor ligands and their apparent preference for different cell types of blood and vasculature has opened up a path to characterize the biological function of GPR15 in greater detail. This should help to prevent adverse effects when the GPR15 receptor–ligand axis is the focus for therapeutic purposes.

## Figures and Tables

**Figure 3 ijms-22-10824-f003:**
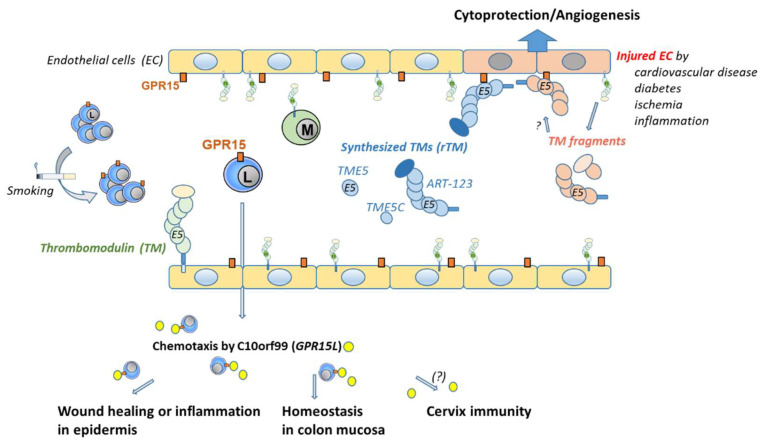
Distribution and binding of GPR15 by ligands in a blood vessel. GPR15 is expressed on endothelial cells (ECs) and on subtypes of lymphocytes. The frequency of GPR15+ lymphocytes can be increased by chronic cigarette smoking. If and how the GPR15 level on ECs can be influenced remains elusive. There are two different ligands for GPR15. One ligand, named C10orf99, is linked to binding on lymphocytes. It is found outside the vessel mainly in the skin, colon, and cervix. C10orf99 is responsible for chemotaxis of GPR15+ lymphocytes that are involved in wound healing or inflammation in the epidermis or homeostasis in the colon mucosa. Its role in cervix mucosa remains elusive. The second ligand is linked to thrombomodulin (TM) and ECs. TM is ubiquitously expressed on the luminal surface of ECs. Synthesized recombinant soluble fragments of TM (rTM), such as ART-123, TME5, and TME5C, were found to bind GPR15. rTM binds to GPR15 via its C-loop of the fifth epidermal growth factor-like region. TM expression decreases upon EC injury [30] and fragments are found in plasma. Binding of these natural fragments to GPR15 could be assumed but has not been confirmed. Binding of TM to GPR15 on ECs is cytoprotective and proangiogenic. *ART-123* recombinant human soluble thrombomodulin, *C10orf99* chromosome 10 open reading frame 99, *EC* endothelial cells, *GPR15* orphan G protein-coupled receptor 15, *L* lymphocyte, *M* monocyte, *TM* thrombomodulin, *TME5* fifth epidermal growth factor-like region of TM, *TME5C* C-loop of TME5.

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
