# Peer review of "The Role of GPR15 Function in Blood and Vasculature"

_ijms, 2021, doi:10.3390/ijms221910824_

Round 1

Reviewer 1 Report

This is an encyclopedic recitation of the literature regarding GPR15 in the circulatory system by a highly respected member of the scientific community. There are a number of strengths and several weaknesses to the manuscript. First off, I am fairly familiar with the subject and I was extremely impressed by the depth and scholarship of the review. It is clear that the author left no stone unturned in his effort to provide the reader with the relevant literature.  Still, the author’s comprehensive review of the studies of GPR15 function turned up quite a few papers with which I was completely unfamiliar. I particularly enjoyed his delineation of the potential ligands for the receptor. Finally, the illustrations are top-notch.  Overall, I am fairly persuaded that this is a piece of literature that needs to be published. Furthermore, I am sure that my colleagues and I will both re-read it and cite it when it is finished.  However, before then, there are a couple things that clearly need to be addressed.

  1. The language in the manuscript needs to be carefully edited by someone more fluent in English. This is a potentially very nice review. I must say I enjoyed reading much of it. However, parts of it were painful to read because of the grammatical issues. I’m quite familiar with Dr. Bauer’s work and this is atypical. For example, with regard to the first sentence, it should read something like this “GPR15 is a member of the Class A orphan G protein-coupled receptor (GPCR) family.”  The sentence that follows is equally clumsy.  Did Dr. Bauer mean to state that “The role of GPR15 in smoking has been recently reviewed”?  Normally, I would attempt to identify specific examples in the text but there are just way too many.  May I suggest that a colleague edit the manuscript?  The material is truly outstanding.  It may take someone 4-8 hours, but the effort would be worth it.  It would make the paper far more readable. 
  2. The title needs to be changed. “Diversity” is a hot button word in English that should be avoided.  Furthermore, the noun doesn’t really describe what the paper covers.  According to the dictionary, there are two definitions: 1) “the state of being diverse; variety” and 2) “the practice or quality of including or involving people from a range of different social and ethnic backgrounds and of different genders, sexual orientations, etc.”  May I suggest “The Role of GPR15 in Blood and Vasculature”?
  3. I really liked the “Open Questions” portion of the manuscript. I think that the authors should expand this section to include a brief synopsis on how enhanced GPR15 may moderate the effects of smoking on the vasculature.  Smoking exerts a lot of its undue effects via the vasculature.  But are these effects mediated by GPR15? I realize Koks et al. covered it in 2017, but they do not own the subject and I would like to see what Dr. Bauer thinks.

Author Response

To 1. The manuscript has been thoroughly edited by a native speaker.

To 2. I fully agree with the recommendation to change the title and changed the title as suggested. Additionally I avoided, as recommended, the word “diversity” in the manuscript. Thank’s for that comment.

To 3. This is a further interesting open question I added under (vi) as such in the paragraph, but not with the focus exclusively on smoking. For smokers, possessing a higher risk for atherosclerosis, more of interest would be whether alternatives, such as dietary compounds, can be found to promote cytoprotective effects on endothelial cells. For patients suffering from disseminated intravascular coagulopathy (DIC) GPR15-ligand works well and do not need apparently an enhanced GPR15 expression on endothelial cells. However, for the large number of smokers the strategy with therapeutics is not the best option.

Reviewer 2 Report

This review discusses recent findings concerning the diversity of GPR15 function in blood and vasculature. 

The review is written very well. All aspects of GPR15 (physiology, pathophysiology, activity regulation) with respect to blood and vasculature were addressed. The latest literature on the topic is discussed and cited. Clearly arranged cartoons illustrate the facts.

I would recommend to consider this manuscript for publishing after minor revision.

  1. the reference list should be revised according to the style of the journal
  2. the font size in Figure 3 must be adapted to the font sizes in Figure 1 and 2

Author Response

To 1. References has been corrected

To 2. The font size has been adapted.